# Balancing model-based and memory-free action selection under competitive pressure

**Atsushi Kikumoto, Ulrich Mayr\***

Department of Psychology, University of Oregon, Eugene, United States

**Abstract** In competitive situations, winning depends on selecting actions that surprise the opponent. Such unpredictable action can be generated based on representations of the opponent's strategy and choice history (model-based counter-prediction) or by choosing actions in a memory-free, stochastic manner. Across five different experiments using a variant of a matching-pennies game with simulated and human opponents we found that people toggle between these two strategies, using model-based selection when recent wins signal the appropriateness of the current model, but reverting to stochastic selection following losses. Also, after wins, feedback-related, mid-frontal EEG activity reflected information about the opponent's global and local strategy, and predicted upcoming choices. After losses, this activity was nearly absent—indicating that the internal model is suppressed after negative feedback. We suggest that the mixed-strategy approach allows negotiating two conflicting goals: 1) exploiting the opponent's deviations from randomness while 2) remaining unpredictable for the opponent.
DOI: https://doi.org/10.7554/eLife.48810.001

## Introduction

Even the most powerful backhand stroke in a tennis match loses its punch when the opponent knows it is coming. Competitions that require real-time, fast-paced decision making are typically won by the player with the greatest skill in executing action plans *and* who are able to choose their moves in the least predictable manner (*Camerer et al., 2015*; *Nash, 1950*; *Morgenstern and Von Neumann, 1953*; *Lee, 2008*). Yet, how people can consistently achieve the competitive edge of surprise is not well understood. The fundamental challenge towards such an understanding lies in the fact that our cognitive system is geared towards using memory records of the recent selection history to exploit regularities in the environment. However, as suggested by decades of research (*Wagenaar, 1972*; *Baddeley, 1966*; *Rapoport and Budescu, 1997*; *Arrington and Logan, 2004*; *Mayr and Bell, 2006*), these same memory records will also produce constraints on current action selection that can work against unpredictable behavior.

One such memory-based constraint on unpredictable action selection is that people often tend to repeat the last-executed action plan. A considerable body of research with the 'voluntary task-switching' paradigm (*Arrington and Logan, 2004*; *Mayr and Bell, 2006*) has revealed a robust per-severation bias, even when subjects are instructed to choose randomly between two different action plans on a trial-by-trial basis—a regularity that in competitions could be easily exploited by a percep-tive opponent.

Another important constraint is the win-stay/lose-shift bias, that is a tendency to repeat the most recently reinforced action and abandon the most recently punished action. Reinforcement-based action selection does not require an internal representation of the task environment and is therefore often referred to as 'model-free'. Previous work has revealed that reinforcement learning can explain some of the choice behavior in competitive situations (*Cohen and Ranganath, 2007*; *Erev and*

**\*For correspondence:**
mayr@uoregon.edu

**Competing interests:** The authors declare that no competing interests exist.

**eLife digest** The best predictor of future behavior is past behavior, so the saying goes. And studies show that in many situations, we do have a tendency to repeat whatever we did last time, particularly if it led to success. But while this is an efficient way to decide what to do, it is not always the best strategy. In many competitive situations – from tennis matches to penalty shoot-outs – there are advantages to being unpredictable. You are more likely to win if your opponent cannot guess your next move.

Based on this logic, Kikumoto and Mayr predicted that in competitive situations, people will toggle between two different decision-making strategies. When they are winning, they will choose their next move based on their beliefs about their opponent's strategy. After all, if your opponent in a tennis match has failed to return your last three backhands, it is probably worth trying a fourth. But if an action no longer leads to success, people will switch tactics. Rather than deciding what to do based on their opponent's strategy and recent behavior, they will instead select their next move more at random. If your tennis opponent suddenly starts returning your backhands, trying any other shot will probably produce better results.

To test this prediction, Kikumoto and Mayr asked healthy volunteers to play a game against real or computer opponents. The game was based on the 'matching pennies' game, in which each player has to choose between two responses. If both players choose the same response, player 1 wins. If each player chooses a different response, player 2 wins. Some of the opponents used response strategies that were easy to figure out; others were less predictable. The results showed that after wins, the volunteers' next moves reflected their beliefs about their opponent's strategy. But after losses, the volunteers' next moves were based less on previous behaviors, and were instead more random. These differences could even be seen in the volunteers' brainwaves after win and loss trials.

As well as providing insights into how we learn from failures, these findings may also be relevant to depression. People with depression tend to switch away from a rationale decision-making strategy too quickly after receiving negative feedback. This can lead to suboptimal behavior patterns that make it more difficult for the person to recover. Future studies should explore whether the short-term decision-making strategies identified in the current study can also provide clues to these behaviors.

DOI: https://doi.org/10.7554/eLife.48810.002

*Roth, 1998*; *Lee et al., 2012*). Yet, players who rely on reinforcement-based selection can also be counter-predicted by their opponent, or run the risk of missing regularities in their opponents' behavior. Therefore, recent research indicates that when performing against sophisticated opponents, model-free choice can be replaced through model-based selection, where choices are based on a representation of task-space contingencies (*Gläscher et al., 2010*), including beliefs about the opponent's strategies (*Donahue et al., 2013*; *Tervo et al., 2014*).

Model-based selection is consistent with the view of humans as rational decision makers (*Nash, 1950*; *Morgenstern and Von Neumann, 1953*), yet also has known limitations. For example, it depends on attentional and/or working memory resources that vary across and within individuals (*Otto et al., 2013a*). In addition, people are prone to judgement and decision errors, such as the confirmation bias, that get in the way of consistently adaptive, model-based selection (*Abrahamyan et al., 2016*).

In light of the shortcomings of both , model-free choice and model-based strategies it is useful to consider the possibility that in some situations, actors can choose in a memory-free and thus stochastic manner (*Donahue et al., 2013*; *Tervo et al., 2014*). Memory-free choice would establish a 'clean-slate' that prevents the system from getting stuck with a sub-optimal strategy and instead allows exploration of the full space of possible moves. Moreover, it reduces the danger of being counter-predicted by the opponent (*Walker and Wooders, 2001*; *Chiappori et al., 2002*). At the same time, an obvious drawback of stochastic choice is that without a representation of the opponent, systematic deviations from randomness in the opponent's behavior remain undetected and therefore cannot be exploited. In addition, just as is the case for model-based selection, stochastic selection puts high demands on cognitive control resources (*Baddeley et al., 1998*) and therefore it is not

clear under which circumstances people can consistently ignore or suppress context representations in order to choose in a memory-free manner (*Rapoport and Budescu, 1992*).

As the model-based and the memory-free strategy both come with strengths and limitations, one potential solution is that people use a simple heuristic to move back and forth between these two modes of selection. Specifically, positive feedback (i.e., wins on preceding moves) could serve as a cue that the current model is adequate and should be maintained. In contrast, negative feedback might serve as a signal that the current model needs to be suspended in favor of a memory-free mode of selection that maximizes exploration and unpredictability.

In the current work, we used an experimental paradigm that provides a clear behavioral signature of model-based versus memory-free choices as a function of preceding win versus loss feedback. We found that following win feedback, people tended to choose their next move both on the basis of recent history and a more global model of the opponent. However following losses, we did not simply see choice behavior revert back towards simple memory-driven biases. Rather choices were less determined by recent history and task context—in other words more stochastic. In addition, we present neural evidence that loss feedback literally 'cleans the slate' by temporarily diminishing the representation of the internal model (*Donahue et al., 2013*; *Tervo et al., 2014*; *Kolling et al., 2016a*).

## Results

### Overview

Our experimental situation marries the voluntary task-switching paradigm, where people switch between action rules, with a two-person, matching-pennies game. Different from the standard matching-pennies game, winning a trial was defined by whether or not players chose matching action rules, rather than simple response choices. Specifically, on each trial players saw a circle, either on the bottom or the top of a vertically arranged frame (see *Figure 1a*). Participants chose on each trial between the 'freeze' rule, which keeps the circle at the same location, and for which responses had to be entered at two vertically arranged keys on the right side of the keyboard, or the 'run' rule, which moves the circle to the opposite end of frame, and for which responses had to be entered at two vertically arranged keys on the left side. On a given trial, for the freeze rule, participants had to press the one right-hand key that was spatially compatible to the current circle location; for the run rule they had to press the one left-hand key that was incompatible with the current

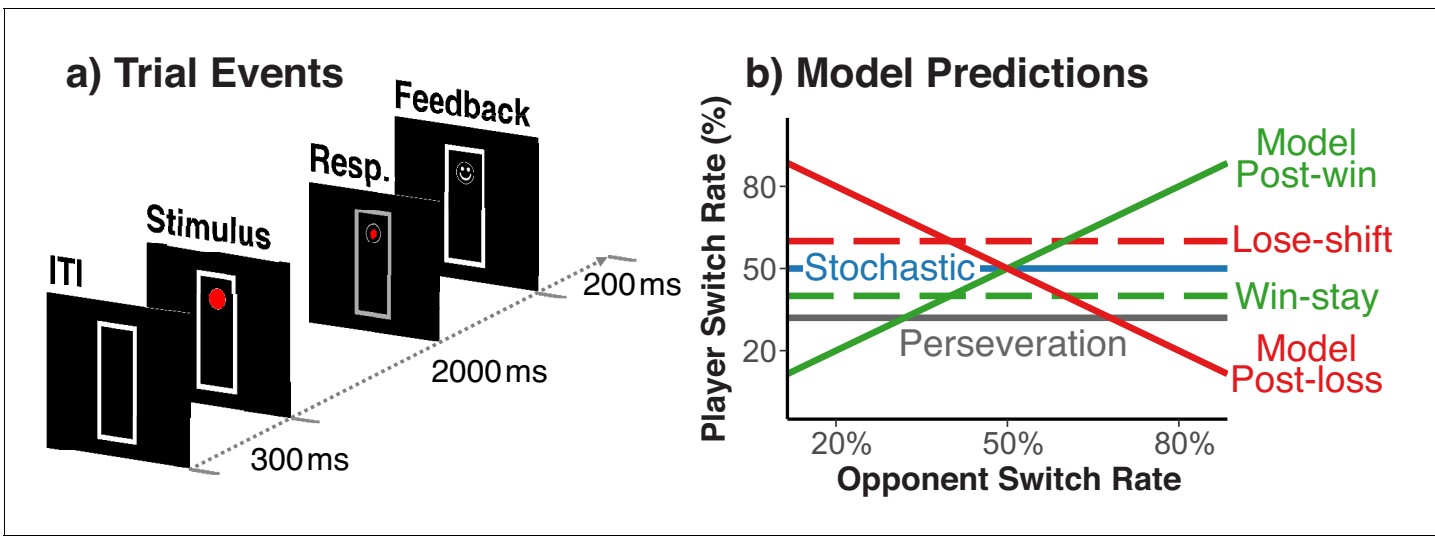

**Figure 1.** Trial events and theoretically possible switch-rate patterns. (**a**) Sequence of trial events and response rules in the fox/rabbit paradigm. (**b**) Idealized predictions of how difference choice strategies and biases are expressed in the player's switch rate. Choices based on an internal model of the opponent, lead to a positive relationship.
DOI: https://doi.org/10.7554/eLife.48810.003

circle position. Wins versus losses were signaled through a smiley or frowny face at the place of the post-response circle position. Players were assigned either the role of the 'fox' or the 'rabbit'. Foxes win a given trial when they 'catch the rabbit', that is when they pick the same rule as the rabbit on that trial. Rabbits win when they 'escape the fox', that is when they pick the rule not chosen by the fox.

We exposed each player to a set of simulated opponents that differed in their average of switching rules from one trail to the next (e.g., 20%, 35%, 50%, 65%, and 80%). Otherwise, these simulated opponents made choices randomly, and did not respond in any manner to the player's behavior. Variations in opponents' switch rate provide a diagnostic indicator of both model-based and stochastic behavior (*Figure 1b*). Specifically, a model-based agent should appreciate the fact that when playing against an opponent who switches frequently between run and freeze rules (i.e., $p>0.5$), it is best to switch rules after a win (i.e., 'following along with the opponent'), but to stick with the same rule after a loss (i.e., 'waiting for the opponent to come to you'); the opposite holds for opponents with a low switch rate (i.e., $p<0.5$). Thus, model-based behavior would produce a combination of the filled green and red switch-rate functions in *Figure 1b*. In contrast, a memory-free agent should produce random behavior (i.e., a switch rate close to $p=0.5$) irrespective of the opponent's strategies (i.e., the blue function in *Figure 1b*). Thus, our hypothesis of a feedback-contingent mix between model-based and stochastic behavior predicts an increase of players' switch rate as a function of their opponents' switch rate for post-win trials (the filled green function in *Figure 1b*), but a switch rate close to $p=0.5$ irrespective of the opponent's switch rate on post-loss trials (the blue function in *Figure 1b*). *Figure 1b* also shows how the lower-level perseveration bias and the win-stay/lose-shift bias would affect the data pattern.

In our rule-selection version of the matching-pennies game, each rule is associated with two specific separate response options, only one of which is 'allowed' for the currently chosen action rule (i.e., 'freeze' vs. 'run' rule). This enabled us to determine if an increase in stochasticity is specific to the generation of action choices, or alternatively due to an unspecific increase of information-processing noise. In the latter case, greater choice stochasticity should go along with more action errors. In this paradigm, participants can make two types of such errors: They can either fail to pick the set of response keys that is consistent with a chosen rule (e.g., right side keys when the intended rule is 'run'), but then execute the response that is consistent with the intended rule (e.g., incompatible response on the right side), or they could correctly pick the side that is consistent with the intended rule, but then execute the wrong response option (e.g., a compatible response on the left side). Without knowing subjects' intended choice on a given trial, we cannot distinguish between errors types. However, either one of these can be interpreted as an action error that occurs independently of the choice between rules. If stochasticity affects information processing in an unspecific manner then we should find that such errors covary with choice stochasticity, both across conditions and across subjects. We also wanted to ensure that our main conclusions are not limited to the rule-selection paradigm. Therefore, we attempted to replicate our basic pattern of results in Experiment three in a standard matching-pennies paradigm with simple response choices (but no way of distinguishing choice stochasticity from unspecific information-processing stochasticity).

## Analytic strategy for testing main prediction

To test the prediction of loss-induced stochastic behavior, we cannot simply contrast the slopes of post-win and post-loss switch-rate functions. Such a comparison would not differentiate between a pattern of post-loss and post-win functions with the same slope but opposite signs (as would be consistent with the model-based choice strategy, see *Figure 1b*) and the predicted pattern of more shallow slopes following losses. Therefore, as a general strategy, we tested our main prediction by comparing slopes after selectively inverting the labels for the opponent switch-rate in the post-loss condition (e.g., 80% becomes 20%). This allows direct comparisons of the steepness of post-win and post-loss switch-rate functions. In the SI, we also present results from standard analyses.

## Modeling choice behavior

Our behavioral indicator of a mix between model-based and stochastic behavior is expressed in players' switch rate, which can also be affected by the perseveration and win-stay/lose-shift bias (see *Figure 1b*). In standard sequential-decision paradigms it is difficult to distinguish between stochastic

behavior and low-level choice biases. Therefore, we attempted to obtain a realistic characterization of the various influences on choice behavior by using a simple choice model to predict the probability of switch choices $p_{switch}$:

$$P_{switch} = \frac{\exp(os * (ms - (wl + 1) * .5 * sm) - pe + wl * -ss)}{1 + \exp(os * (ms - (wl + 1) * .5 * sm) - pe + wl * -ss)} \quad (1)$$

with: $os = \ln(p_{os}/ (1- p_{os}))$; post-win: $wl = 1$, post-loss: $wl = -1$;

where $p_{os}$ is the opponent's switch rate, which is translated into its log-odds form ($os$); $wl$ codes for wins versus losses on trial $n$-1. The parameter $ms$ (**m**odel **s**trength) represents the strength of the model of the opponent ($ms = 1$ would indicate direct probability matching between the opponent's and the player's switch probability). The parameter $sm$ (**s**trategy **m**ix) represents the degree to which the model-based choice is changed on post-loss relative to post-win trials; a negative $sm$ parameter would indicate suppression of the model in favor of stochastic choice following losses. In addition, a positive $pe$ (**pe**rseveration **e**ffect) parameter represents the tendency to unconditionally favor the previously selected choice, and a positive $ss$ (win-**s**tay/lose-**s**hift) parameter expresses the strength of the win-stay/lose-shift bias. We present predictions from this model in *Figure 2*, and report additional details of the modeling results in section *Modeling Results*.

## Choice behavior with simulated opponents

*Experiment 1*. In this experiment, we establish the basic paradigm. As shown in the upper-left panel of *Figure 2*, participants increased their switch rate as a function of their opponents' switch rates following win trials. In contrast, on post-loss trials, the change in players' switch rate (as a function of their opponents' switch rate) was much smaller than on post-win trials and it was centered at $p$=0.5—a pattern that is consistent with largely stochastic choice. The condition with an opponent switch rate of $p$=0.5 most closely resembles previous studies that have reported a win-stay/lose-shift bias in competitive situations (*Cohen and Ranganath, 2007*). In fact, for this condition, we did find a significantly higher switch rate after losses than after wins, indicating that reward-based choices are one factor that affects choice.

Following win trials, participants' switch rate follows opponents' switch rate when the opponents' switch rate was low, but only in a muted manner when the opponents' switch rate was high (i.e., the switch-rate function was less than 1.0). We attribute this reluctance to fully endorse the model-based strategy to the influences of counteracting, lower-level, win-stay/lose-shift and perseveratory tendencies. Indeed, as will be described in greater detail in the section *Modeling Results*, results from applying our choice model to the data indicate that (a) a strong tendency towards model-based choices on post-win trials, (b) an increase of stochastic choice on post-loss trials, (c) a general perseveratory tendency, and (d) a win-stay/lose-shift bias all contribute to the overall choice behavior. The *Figure 2—figure supplement 1–3* provide additional information about determinants of choice behavior and also of participants' success rate.

*Experiment 2*. Following feedback from the previous trial, participants had only 300 ms to choose their move for the next trial in Experiment 1. Therefore, one might argue that the observed stochastic choice is simply a result of negative feedback temporarily interfering with model-based selection (*Otto et al., 2013a*). To examine this possibility, we manipulated the inter-trial-interval (ITI) in Experiment 2 between 300 ms and 1000 ms. As shown in *Figure 2*, this manipulation had no effect, indicating that stochastic choice is not due to loss-induced processing constraints.

*Experiment 3*. The fox/rabbit task was modeled after the voluntary task-switching to allow us to distinguish between choice stochasticity and more general increase of noise in the cognitive system. However, it is important to explore how the observed pattern might change with less complex response rules than used in this paradigm. We therefore implemented in Experiment 3 simple choices without any contingencies on external inputs (i.e., the fox wins when selecting the same up or down location as the rabbit, and vice versa). Here, we generally found a stronger expression of model-based choice following *both* losses and wins, and also much less perseveration bias (*Figure 2*). Yet the win-loss difference in switch-rate slopes remained just as robust as in the other experiments. Thus, the more complex actions that players had to choose from in Experiments 1 and 2 may have suppressed the overall degree of model-based action selection (*Otto et al., 2013b*). However,

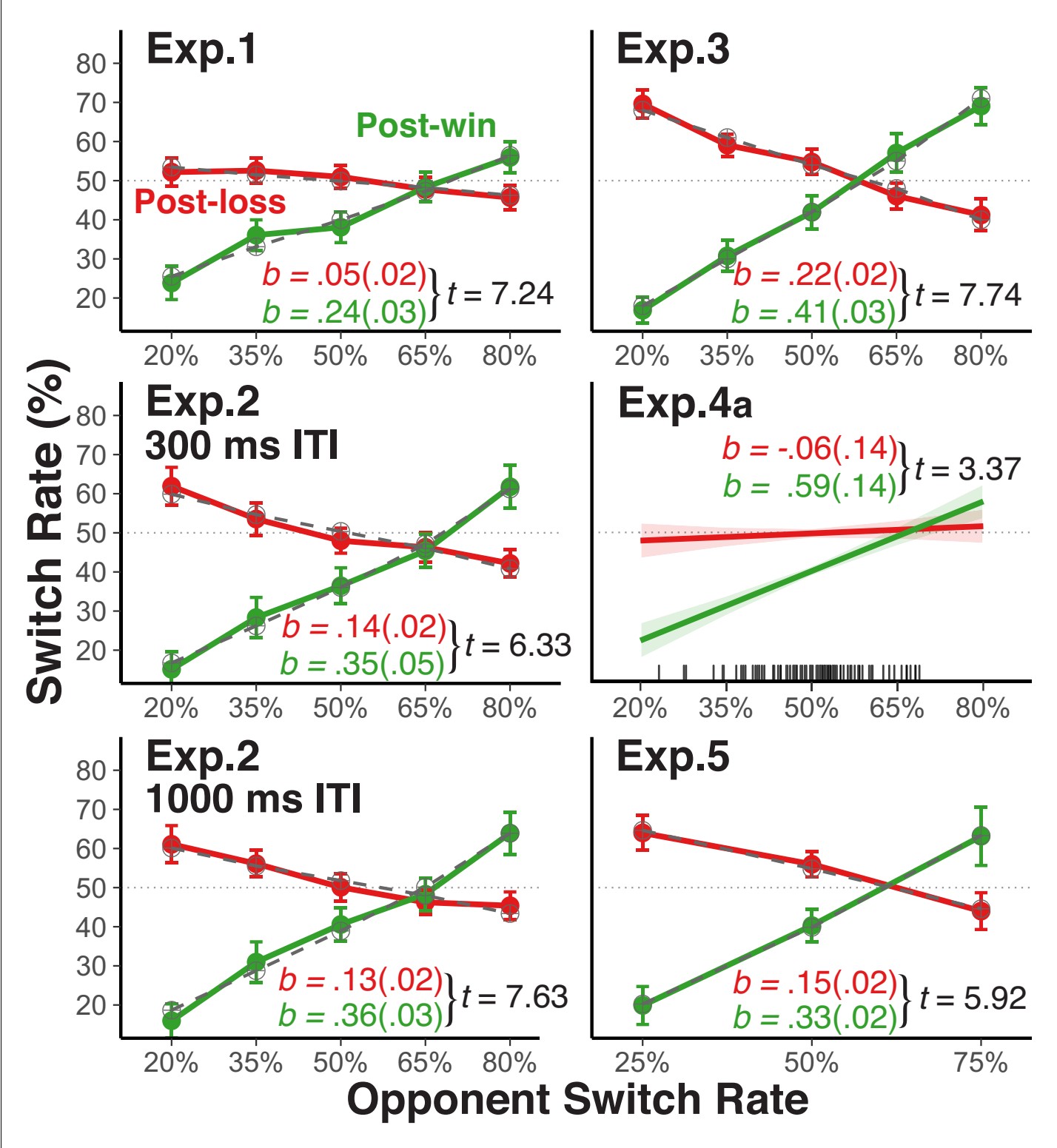

**Figure 2.** Player's average switch rates as a function of opponents' switch rates. Average empirical switch rates for post-win and post-loss trials as a function of the simulated opponents' switch rates for Experiment 1, 2, 3, and five and the average switch rate of each human opponent in Experiment 4a (tick marks on the x-axis indicate individual average switch rates). The dashed lines for Experiment 1, 2, 3, and 5 show the predictions of the theoretical choice model applied to the group average data (see sections *Modeling Choice Behavior* and *Modeling Results*). Error bars represent 95% within-subject confidence intervals. For the analyses, we regressed the player's switch rate on the opponent's switch rates, the win-loss contrast, and the interaction between these two predictors after reversing the labels of the opponents' switch rate predictor for post-loss trials (see section *Analytic*

*Figure 2 continued*

*Strategy for Testing Main Prediction*). As a test of these interactions, we show the corresponding *t*-values (*SE*): the unstandardized slope coefficients (*SE*; green = post win, red = post loss) were derived from separate analyses for post-win and post-loss trials.

DOI: https://doi.org/10.7554/eLife.48810.004

The following figure supplements are available for figure 2:

**Figure supplement 1.** Are feedback effects temporary?

DOI: https://doi.org/10.7554/eLife.48810.005

**Figure supplement 2.** Rate of winning as a function of opponent switch rate and n-1 wins/losses.

DOI: https://doi.org/10.7554/eLife.48810.006

**Figure supplement 3.** Analysis of action choices.

DOI: https://doi.org/10.7554/eLife.48810.007

**Figure supplement 4.** Switch rates when competing versus not competing.

DOI: https://doi.org/10.7554/eLife.48810.008

response rule complexity did not appear to affect the win/loss-contingent difference in the relative emphasis on model-based versus stochastic choices.

## Competition against human players

*Experiment 4a*. It is possible that the observed pattern of results is specific to experimental situations with a strong variation in simulated, opponent switch rates. To examine the degree to which this pattern generalizes to a more natural, competitive situation, we used in Experiment 4a pairs of participants who competed with each other in real time, with one player of each dyad acting as fox, the other as rabbit (see *Figure 2—figure supplement 4* for a comparison between the competitive Experiment 4a and the non-competitive Experiment 4b). Obviously, the naturally occurring variation in switch rates was much lower than in the experiments using simulated opponents (see distributions of individuals average switch rates in *Figure 2*). Nevertheless, the estimated slopes linking players' switch rates to opponents' switch rates showed a very similar pattern as in the other experiments with simulated opponents.

## Modeling results

We applied our choice model both to the group-average switch rates for each experiment, and to the individuals-specific switch rates. *Table 1* shows the estimated parameters for each of the four experiment with simulated opponents, as well as model fits ($R^2$) for the group-level data. We found that each of the four different factors (i.e., model strength, suppression of model/strategy mix, perseveration effect, and win-stay/lose-shift bias) were relevant for characterizing participants' choices.

**Table 1.** Parameter estimates and 95% confidence intervals from fitting the choice model to group average and individual data from Experiments 1, 2, 3, and 5.

| Parameters | Fitting group averages | | | | | Fitting individuals' Data | | | |
|---|---|---|---|---|---|---|---|---|---|
| | *ms* | *sm* | *pe* | *ss* | $R^2$ | *ms* | *sm* | *pe* | *ss* |
| *Simulated Opp.* | | | | | | | | | |
| Exp. 1 | 0.48 ± 0.10 | −0.38 ± 0.09 | 0.21 ± 0.06 | 0.20 ± 0.06 | 0.975 | 0.61 ± 0.19 | −0.50 ± 0.16 | 0.24 ± 0.14 | 0.22 ± 0.11 |
| Exp. 2, 300 ms | 0.74 ± 0.12 | −0.47 ± 0.15 | 0.28 ± 0.07 | 0.30 ± 0.07 | 0.988 | 0.96 ± 0.26 | −0.66 ± 0.24 | 0.34 ± 0.10 | 0.36 ± 0.14 |
| Exp. 2, 1000 ms | 0.74 ± 0.13 | −0.49 ± 0.17 | 0.19 ± 0.08 | 0.26 ± 0.08 | 0.984 | 0.93 ± 0.24 | −0.67 ± 0.21 | 0.25 ± 0.11 | 0.33 ± 0.13 |
| Exp. 3 | 0.86 ± 0.11 | −0.44 ± 0.14 | 0.07 ± 0.04 | 0.24 ± 0.06 | 0.993 | 1.13 ± 0.28 | −0.68 ± 0.23 | 0.11 ± 0.10 | 0.29 ± 0.11 |
| Exp. 5 | 0.87 ± 0.17 | −0.50 ± 0.21 | 0.11 ± 0.09 | 0.30 ± 0.09 | 0.998 | 1.10 ± 0.39 | −0.71 ± 0.30 | 0.16 ± 0.10 | 0.36 ± 0.24 |
| *Human Dyads* | | | | | | | | | |
| Exp. 4a | | | | | | 0.16 ± 0.10 | −0.14 ± 0.13 | 0.11 ± 0.9 | 0.31 ± 0.9 |

*Note. ms* = model strength, *sm* = suppression of model (strategy mix), *pe* = perseveration effect, *ss* = win stay/lose-shift tendency. For Experiment 2, fits are reported separately for the 300 ms and the 1000 ms ITI condition. Fits for individual subjects in Experiments 1, 2, 3, and five are on the basis of each subject's condition averages. For Experiment 4a, we report parameters resulting from modeling individuals' trial-by-trial choices.

DOI: https://doi.org/10.7554/eLife.48810.009

For condition-average data, the model strength parameter (*ms*) ranged between. 48 and. 87, indicating that overall, the opponent's switch rate affected the participant's switch rate in an incentive-compatible manner. Average *ms* values below 1.0 indicate that participants overall engaged in 'imperfect' probability matching (*ms* = l0.0 would indicate perfect probability matching; *ms* >1.0 would indicate a maximizing tendency). This pattern is consistent with the previous literature, which suggests that probability matching is the dominant, albeit suboptimal strategy in serial decision tasks (*James and Koehler, 2011*; *Gaissmaier and Schooler, 2008*).

The individual-specific parameter estimates also allowed us to examine the degree to which the different influences on choice were tied to competitive success. To this end, we entered each individual's, four parameter estimates as fixed-effect predictors into a two-level regression analysis with experiment as a random factor and overall success (i.e., probability of win trials) as criterion variable. While on average, *pe* and *ws* indicated the expected perseveration and win-stay/lose-shift biases (i. e., *pe* <0 and *wl* <0), there were substantial individual differences in these parameters, including individuals with alternation or win-shift/lose-stay biases (i.e., *pe* >0 and *ss* >0; *Table 1*). Given that any bias implies a deviation from optimal performance, we coded these two parameters in absolute terms (we obtained similar results with signed values). As shown in *Table 2*, model strength had a highly robust positive effect on success, whereas either a perseveration or an alternation bias reduced the amount of money earned; no corresponding effect was found for the win-stay/lose-shift parameter. As would be expected, the main effect of the strategy-choice parameter was positive, implying that less stochastic behavior after losses produced greater overall success.

In Experiment 4a, participants were paired up to play against each other. Thus, here we had to use the natural, within-session variability of the opponent of each player for the opponent switch rate variable in a trial-by-trial version of our choice model. Therefore, the $p_{os}$ parameter was calculated as a running average of the opponent's switch rate within each block. The latest running average from block *n*-1 was used as a starting value for block *n* (p=0.5 for the first block), allowing some carry-over of prior knowledge of the opponent's previous-block behavior.

Results from this model are shown in the bottom row of *Table 1*. Not surprisingly, the model strength (*ms*) was substantially smaller than in the preceding experiments, but still significantly larger than 0. The perseveration effect (*pe*) and the win-stay/lose-shift bias (*ss*) were roughly in a similar range as in the remaining experiments. Importantly, the suppression of the model parameter (*sm*) was also statistically significant and of about the same size as the model strength parameter (*ms*), indicating that on post-loss trials the effect of the model is essentially eliminated.

We again used a multi-level regression model with participants grouped within dyads to predict each participant's success (in terms of probability of win trials) as a function of the four model parameters. As in the preceding model (*Table 2*), we again used absolute values from the perseveration and the win-stay/lose-shift scores in order to capture biases in in either direction (very similar results would have been obtained with signed values). Results showed greater reliance on the model, a smaller tendency to disregard the model after losses (i.e., a less negative *sm* score), a smaller,

**Table 2.** Using parameter estimates from the choice-model fitted to individual's condition means to predict individual's competitive success.

| | B | SE | t-value |
|---|---|---|---|
| intercept | 0.504 | | |
| *ms* | 0.086 | 0.007 | 12.31 |
| *sm* | 0. 064 | 0.008 | 8.03 |
| abs(*pe*) | −0.016 | 0.006 | −2.56 |
| abs(*ss*) | −0.001 | 0.001 | −0.10 |

*Note.* Shown are fixed-effect coefficients (*b*), the standard error around the coefficients (SE), and the associated *t*-value. Experiment was coded as a random grouping factor. Absolute values for the *pe* and the *wl* effect were used to account for biases in either direction. Note, that the more negative the *sm* parameter, the greater the suppression of model-based choice on post-loss trials. Thus, a positive coefficient in this analysis indicates that less suppression leads to higher earnings.

DOI: https://doi.org/10.7554/eLife.48810.010

absolute perseveration score, and a larger absolute win-stay/lose-shift bias all contributed to greater success (*Table 3*). Aside from the somewhat surprising result for the win-stay/lose-shift parameter, the overall pattern was qualitatively very similar to the results from the simulated-opponent experiments.

Combined, the modeling results reveal that choices are influenced by low-level influences (perseveration and win-stay/lose-shift bias) as well by a model of the opponent's strategy. Most importantly, we found that over and above these previously established influences, the parameter reflecting a loss-contingent reduction of the model had a robust influence on choice behavior. Moreover, the different parameters had distinct effects on individual differences in competitive success, with the loss-contingent reduction of the model (i.e., increase in stochasticity) clearly representing a sub-optimal influence.

## RT and error effects on model-based and memory-free choices

As described in the overview section, different from standard choice paradigms (*Daw et al., 2006*; *Muller et al., 2019*), the current paradigm allows us to distinguish between stochasticity during the choice between action rules and general information-processing noise (*Kane et al., 2017*). If the loss-induced choice stochasticity is due to a general increase in information processing noise then we should see that greater stochasticity goes along with more errors and possibly also with slower responses.

*Figure 3* shows each individual's degree of model-based choice (expressed in terms of absolute switch-rate slopes) after loss and win trials and as a function of both RTs or error rates. In most experiments, there was a slight increase in error rates following loss trials (see marks beneath the x-axis). However, across individuals, the substantial reduction in model-based choice after loss trials was *not* associated with a consistent increase in error rates or RTs. Likewise, in multilevel regression models with the absolute switch-rate slopes as dependent variable, the post-win/loss contrast remained highly robust after controlling for RTs and errors as within-subject fixed effects (range of $t$-values associated with the post-win/loss predictor: 3.96-10.78).

## Loss-induced increase of stochastic choice

So far, we have established that participants were more sensitive to their opponents' global strategies (i.e., the average switch rates) following win than following loss trials. Next, we examined the degree to which these win-loss differences generalized to players' consideration of the recent history of their opponents' and their own choices. To this end, we used multi-level logistic regression models with the switch/repeat choice as criterion. The models included the trial $n$-1 to $n$-3 switch/repeat decisions for opponents and for players, along with the opponents' overall switch rate and were separately run for post-loss and post-win trials to generate the coefficients presented in *Figure 4*.

To directly compare the size of the coefficients, irrespective of their sign, we again reversed the labels, both for the opponents' global switch rate, but also for both the opponent's and the player's $n$-1 to $n$-3 switch/repeat decisions (e.g., switch becomes repeat; see sections *Analytic Strategy for*

**Table 3.** Using parameter estimates from the choice-model fitted to individual's trial-by-trial data to predict the proportion of win trials (n = 94) in Experiment 4a.

|  | B | SE | t-value |
| --- | --- | --- | --- |
| intercept | −0.530 | 0.005 |  |
| *ms* | 0.043 | 0.015 | 2.91 |
| *sm* | 0. 034 | 0.012 | 2.97 |
| abs(*pe*) | −0.080 | 0.019 | −4.11 |
| abs(*ss*) | 0.062 | 0.014 | 4.48 |

*Note.* Shown are the unstandardized regression coefficients (*b*), the standard error around the coefficients (*se*), and the associated *t* and *p* values. Note, that the more negative the *sm* parameter, the greater the suppression of model-based choice on post-loss trials. Thus, a positive coefficient in this analysis indicates that less suppression leads to higher earnings.

DOI: https://doi.org/10.7554/eLife.48810.011

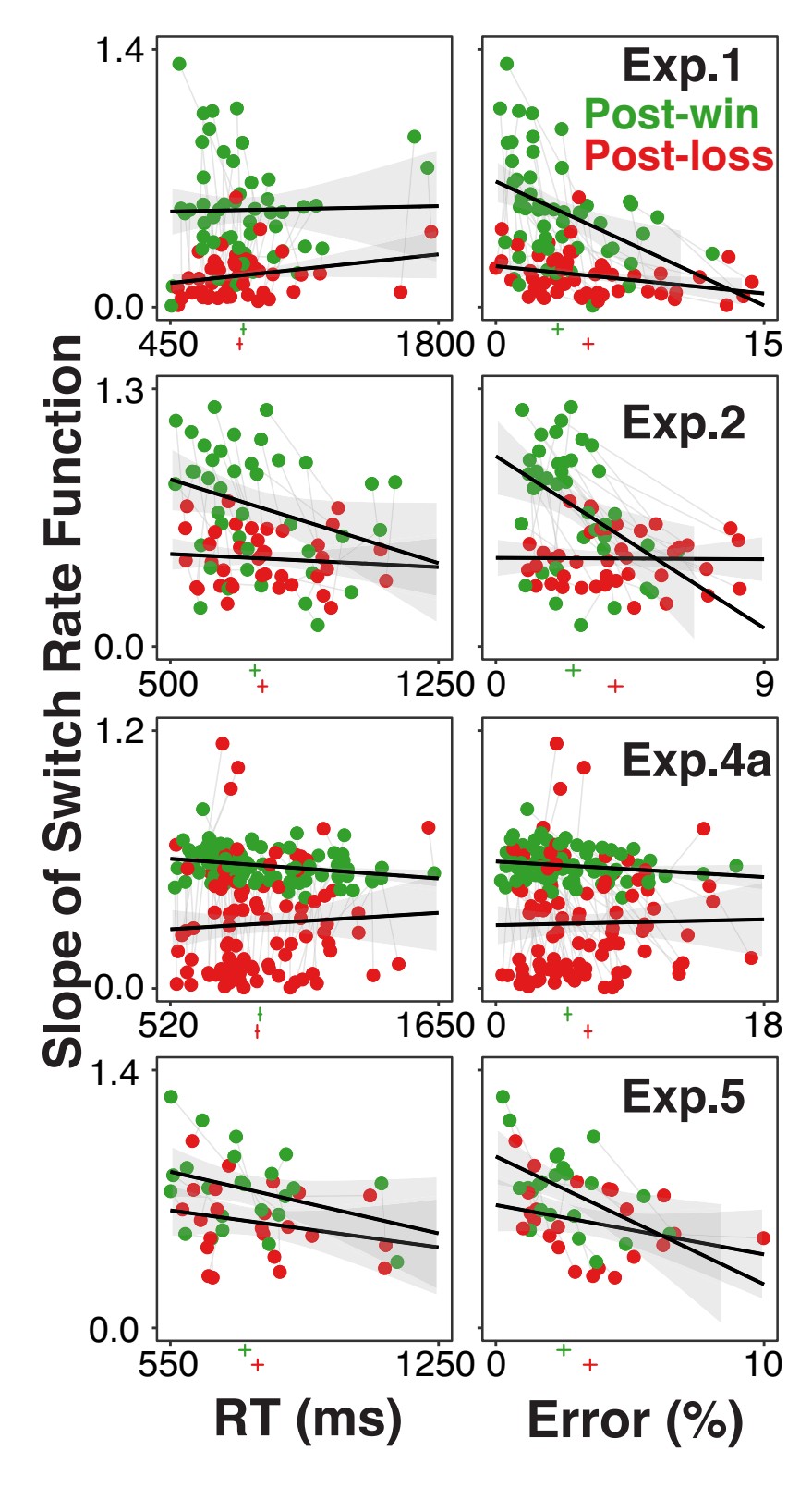

**Figure 3.** Individual participants' degree of model-based choice (indicated by slopes of switch rate functions) in relationship to RTs and errors, separately for post-win (green) and post-loss (red) trials, and for each experiment using the rule-selection paradigm. Each participant is represented both in the post-win and the post-loss condition. The green and red vertical lines below the x-axis of each graph indicate average RTs and error rates, the horizontal lines indicate 95% within-subject confidence intervals. If the increase of choice stochasticity between post-win and post-loss trials were

*Figure 3 continued on next page*

*Figure 3 continued*

due to greater, general information-processing noise, then the win/loss-related decrease in lopes of the switch rate functions would be accompanied by consistent increases in RTs and/or error rates.

DOI: https://doi.org/10.7554/eLife.48810.012

*Testing Main Prediction* and *History Analyses*). Consistent with the prediction that switch/repeat choices following losses are less dependent on recent history, the coefficients for the opponents' history and also the players' own history were in most cases substantially lower after loss than after win trials. In figure supplements to Figure 4, we also show the signed coefficients as well as the results of an alternative analysis that does not require the reversal of labels.

## Neural evidence for memory-free choice following losses

Research with animal models and human neuroimaging work indicates that midfrontal brain regions, such as the anterior cingulate cortex are involved in action-relevant representations and in the gating between different modes of action selection (*Daw et al., 2005*; *Behrens et al., 2007*; *Holroyd and Coles, 2002*). Further, a large body of research suggests that midfrontal EEG activity in response to action feedback contains prediction error signals (*Cohen and Ranganath, 2007*; *Cavanagh et al., 2012*; *Gehring et al., 1993*; *Cavanagh and Frank, 2014*; *Cohen et al., 2011*; *Luft, 2014*), which in turn are reflective of action-relevant expectancies (i.e., the current task model). Therefore, it is theoretically important to link our behavioral results to this broader literature. Specifically, it would be useful to show that (a) only on post-win, but not on post-loss trials, the midfrontal EEG signal contains information about the choice context/model, and (b) that the context information contained in the EEG signal is in fact predictive of upcoming choices.

In Experiment 5, we assessed EEG while participants played the fox/rabbit game against three different types of opponents (25%, 50%, 75% switch rate). The ITI was 700 ms to capture feedback-related EEG signals developing prior to the onset of upcoming trials. The behavioral results were consistent with the other experiments (see *Figures 2*, *3* and *4*).

We conducted a two-step analysis of the EEG signal. In the first step, we tested the prediction that the mid-frontal EEG signal contains less information about the choice-relevant context after loss-feedback than after win-feedback. To this end, we regressed trial-to-trial EEG signals on A) the opponent's overall switch rate, B) the opponent's lag-1 switch/no-switch, C) the player's lag-1 switch/no-switch, and D) the interaction between A) and B), that is between the local and global switch expectancies. The latter term was included to capture the fact that if feedback-related EEG reflects expectancies about opponents' switch rates, local switch expectancies may depend on the global switch-rate context (*Cavanagh and Frank, 2014*).

The standardized coefficients shown in *Figure 5a* indicate the amount of information about each of the four context variables that is contained in the mid-frontal EEG signal. As apparent, the EEG signal showed a robust expression of the history/context variables following win feedback. Following loss feedback, context information is initially activated, but then appears to be suppressed compared to post-win trials, and trends towards zero at the end of the feedback period. Accordingly, coefficients were significantly larger in post-win trials than in post-loss trials, opponents' overall switch rate: $b = 0.07$, $se = 0.01$, $t(25)=5.22$, p<0.001, opponents' lag-1 switch/no-switch: $b = 0.04$, $se = 0.01$, $t(25)=4.14$, p=0.001, player's lag-1 switch/no-switch: $b = 0.01$, $se = 0.009$, $t(25)=1.19$, p=0.24, interaction between opponents' overall switch rate and lag-1 switch: $b = -0.08$, $se = 0.01$, $t(25)=-7.22$, p<0.001. Given that feedback is related to subject's propensity of switching on the upcoming trial, it is in principle possible that these coefficients simply reflect preparation or increased effort for the upcoming switch. However, as we show in *Figure 5—figure supplement 1*, controlling for the upcoming switch has negligible effects on results. These analyses also show that while there is detectable information about the upcoming switch/no-switch choice, the decodability of the upcoming choice is not modulated by win/loss feedback (see also, 14).

Our analytic strategy deviates from the standard approach of analyzing the EEG signal in terms of feedback-locked, event-related potentials (ERPs; see *Figure 5b*). We used our approach because we did not have a-priori predictions about how exactly the combination of different history/context variables would affect ERPs. More importantly, our regression-based approach naturally yields trial-by-

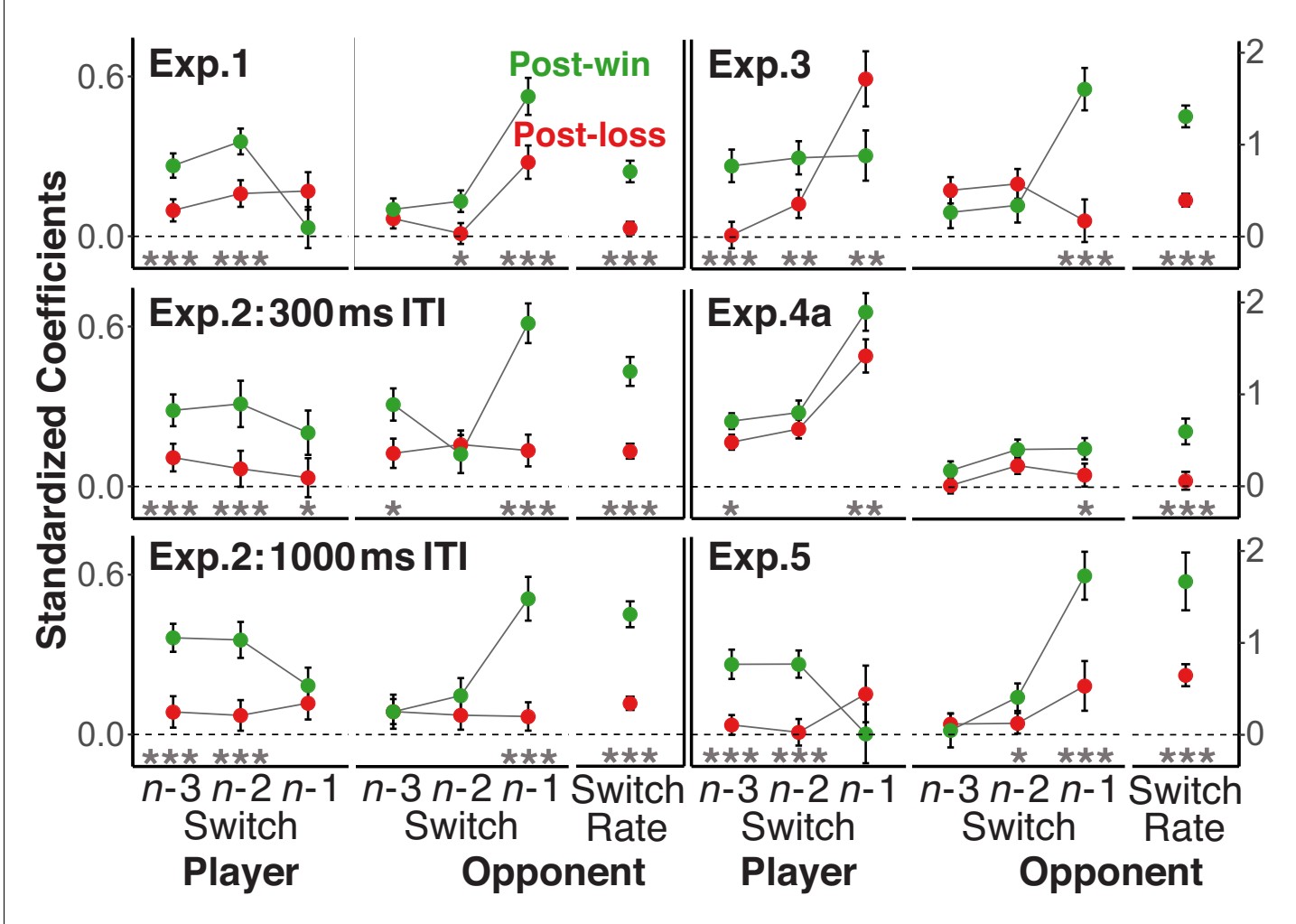

**Figure 4.** Standardized coefficients from multi-level logistic regression models predicting the trial *n* switch/no-switch choice on the basis of players' and opponents' switch/repeat choices on trials *n*-1 to *n*-3 and the opponents' overall switch rate. Error bars are standard errors around the coefficients. To focus on the difference in the strength of relationships rather than their sign, the labels for all opponent predictors were reversed for post-loss trials (see section *Analytic Strategy for Testing Main Prediction*). In addition, we also reversed the labels for all player-related predictors with a win/loss switch in sign. For a statistical test of the size difference between post-win and post-loss coefficients, all history/context variables were included into one model together with the post-win/post-loss contrast and the interaction between this contrast and each of the history/context predictors. Significance levels of the interaction terms are indicated in the figure,<0.05, *<0.01, ***<0.001.

DOI: https://doi.org/10.7554/eLife.48810.013

The following figure supplements are available for figure 4:

**Figure supplement 1.** History analysis with signed action choices.

DOI: https://doi.org/10.7554/eLife.48810.014

**Figure supplement 2.** Alternative analysis of history effects.

DOI: https://doi.org/10.7554/eLife.48810.015

trial indicators of the expression of context-specific information, which can be used in the second step of our analysis (see below), and which would be difficult to obtain through standard ERP analyses. In *Figure 5—figure supplement 3*, we also show that the ERP results are indeed generally consistent with a prediction-error signal that is more strongly modulated by the choice context after wins than losses.

The conclusion that post-loss stochastic behavior occurs because context representations are suppressed, would be further strengthened by evidence that the information contained in the EEG signal is actually relevant for upcoming choices. Therefore, as the second step, we conducted a

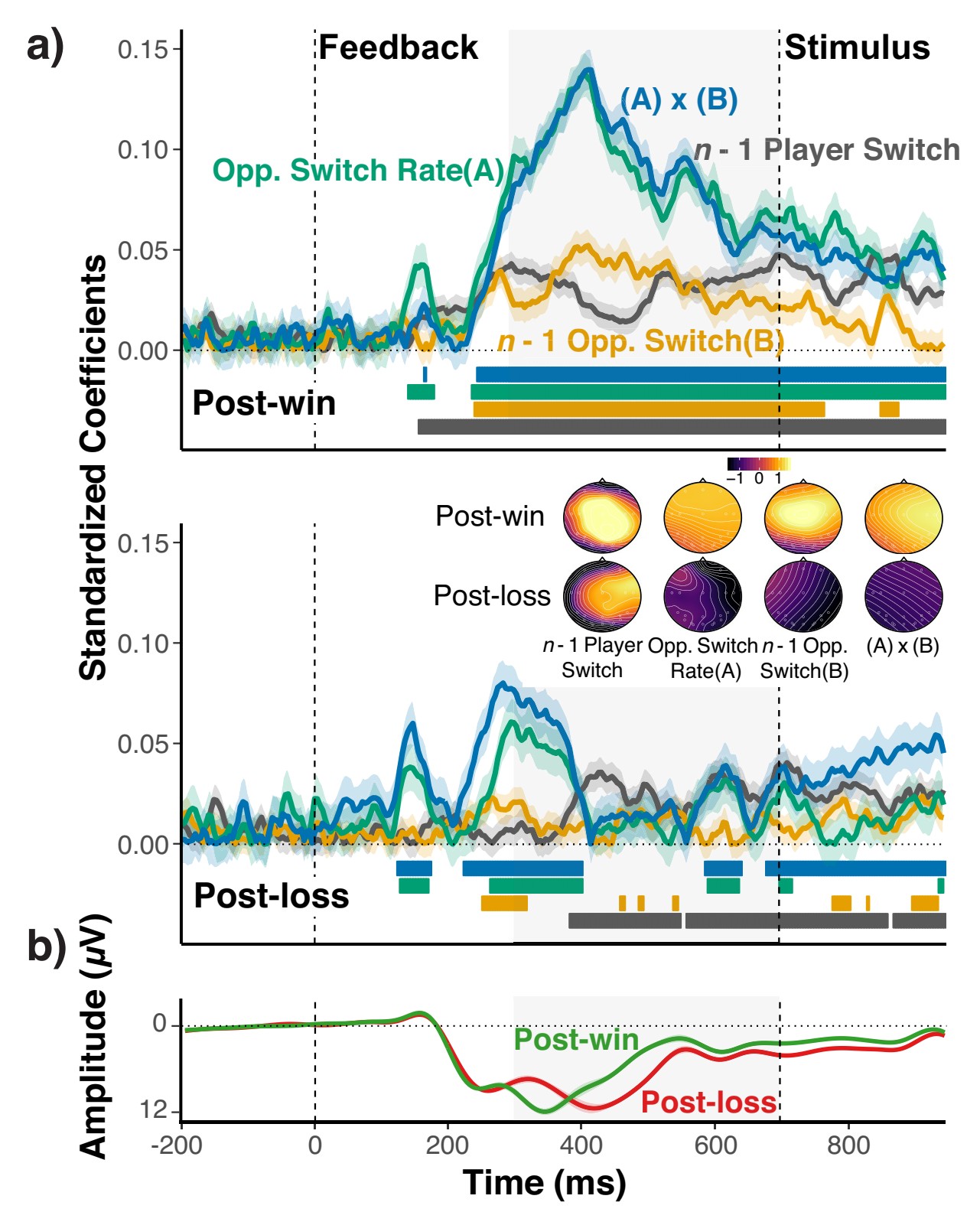

**Figure 5.** EEG-Analysis of choice-relevant information after wins and losses. (a) Standardized coefficients from multi-level regression models relating EEG activity at Fz and Cz electrodes to the opponents' overall switch rate (A), the *n*-1 opponent switch/no-switch choice (B), the *n*-1 players' switch/no-switch choice, and the interaction between (A) and (B) for each time point and separately for post-win (upper panel) and post-loss (lower panel) trials. Shaded areas around each line indicate within-subject standard errors around coefficients. As coefficients for opponent-related predictors

*Figure 5 continued on next page*

*Figure 5 continued*

showed a marked, win/loss flip in sign, we again reversed the label of the post-loss predictors (see section *Strategy for Testing Main Prediction* and *Figures 2* and *3*; for signed coefficients, see *Figure 5—figure supplement 2*). For illustrative purposes, colored bars at the bottom of each panel indicate the time points for which the coefficients were significantly different from zero ($p<0.05$, uncorrected). See text for statistical tests of the predicted differences between coefficients for post-win and post-loss trials. The insert shows the topographic maps of coefficients that result from fitting the same model for each electrode separately. Prior to rendering, coefficients were z-scored across all coefficients and conditions to achieve a common scale. (b) Average ERPs for post-win and post-loss trials, showing the standard, feedback-related wave form, including the feedback-related negativity (i.e., the early, negative deflection on post-loss trials). Detailed ERP results are presented in *Figure 5—figure supplement 3*.

DOI: https://doi.org/10.7554/eLife.48810.016

The following figure supplements are available for figure 5:

**Figure supplement 1.** Controlling for Upcoming Switch and n-1 Stimulus/Response Positions.
DOI: https://doi.org/10.7554/eLife.48810.017
**Figure supplement 2.** Signed predictors.
DOI: https://doi.org/10.7554/eLife.48810.018
**Figure supplement 3.** Event-related potentials.
DOI: https://doi.org/10.7554/eLife.48810.019

psychophysical interaction (PPI) analysis (*Friston et al., 1997*). In a multi-level, logistic regression analysis, we predicted players' trial *n* switch choices, based on 1) the set of four context variables from the preceding analysis for trial *n*-1, 2) the trial *n*-1 residuals from the preceding analysis (reflecting trial-by-trial variations in the EEG signal after controlling for the four context variables), (*Morgenstern and Von Neumann, 1953*) and the corresponding four interactions between the residuals and the context variables. As shown in *Table 4*, we found for post-win trials significant main effects for the residual EEG signal and all context variables. Most importantly, the residual EEG signal modulated how the upcoming choice was affected by the opponent's *n*-1 switch/repeat. These results indicate that the information about context variables contained in the EEG signal is indeed relevant for choices.

Given the reduced context representation following losses (see *Figure 5*), one might expect that there is not sufficient trial-by-trial variability in such information to influence choices. However, it is also possible that even the remaining variability still has predictive power. Therefore, the post-win/ post-loss difference in predicting choices is theoretically less informative than the finding of reduced context representations in the EEG-signal per se and the fact that these representations generally predict upcoming choices. Nevertheless, *Table 4* shows the results also for post-loss trials. The relationship between the EEG representation of the global, opponent switch rate and the upcoming choice was as strong as for post-win. For the local history variables, we found robust effects only

**Table 4.** Coefficients from the PPI analysis predicting upcoming choices using residuals of MLM regression model for post-win and post-loss trials.

| | Post-win | | | Post-loss | | |
|---|---|---|---|---|---|---|
| | *B* | *SE* | *z*-value | *B* | *SE* | *z*-value |
| Opponent Switch Rate (A) | **1.44** | **.047** | **31.15** | **−0.62** | **0.046** | **−13.47** |
| *n*-1 Opponent Switch (B) | **0.76** | **0.038** | **19.96** | −0.07 | 0.034 | −1.99 |
| *n*-1 Player Switch (C) | **0.15** | **0.033** | **4.57** | **−0.23** | **0.033** | **−6.80** |
| A x B | **−0.17** | **0.068** | **−2.57** | 0.08 | 0.061 | 1.37 |
| Residual EEG (D) | **0.20** | **0.03** | **6.39** | **0.07** | **0.04** | **2.05** |
| D x A | **0.11** | **0.05** | **2.26** | **−0.11** | **0.05** | **−2.37** |
| D x B | **−0.13** | **0.04** | **−3.31** | −0.03 | 0.03 | −0.95 |
| D x C | **−0.11** | **0.03** | **−3.36** | −0.01 | 0.03 | −0.36 |
| D x A x B | −0.06 | 0.07 | −0.89 | 0.08 | 0.06 | 1.37 |

*Note.* Shown are the unstandardized regression coefficients (B), the standard error around the coefficients (SE), and the associated z values. Bolded values indicate significant effects (i.e., **z-values > 2**).
DOI: https://doi.org/10.7554/eLife.48810.020

following win trials, but not following loss trials. Note, that the pattern of identical signs for the EEG-behavior relationship across post-win and post-loss trials and the flipped signs for the opponent switch rate/EEG relationship (*Figure 5—figure supplement 2*) is consistent with the reversal of the relationship between opponent, overall switch rate and player switch rate depending on win or loss feedback (e.g., *Figure 2*).

As a final step, we also examined if variations in the strength of history/context representations can account for individual differences in choice behavior. We derived for each individual and predictor, the average, standardized coefficient from the analysis presented in *Figure 5* across the 300 ms to 700 ms interval. Separately for post-win and post-loss trials, we correlated these scores with two behavioral measurements: 1) individuals' switch-rate functions as an indicator for model-based choice (see *Figure 2*) and 2) the overall rate of winning. For post-loss trials, we again used opponent-related predictors with reversed labels (see EEG Recording and Analysis section for details). Thus, for all analyses, more positive scores are indicative of individuals with more model-conform behavior.

As shown in *Figure 6*, coefficients from post-win EEG signals generally predicted the variability among individuals in the degree of model-based adaptation and the rate of winning (except for coefficients of *n*-1 player's switch). In contrast, such relationships were absent for post-loss trials. Here, we also found significant post-win/post-loss differences. For the switch-rate function slopes, post-win/post-loss differences were present for the opponents' lag-1 switch/no-switch contrast, $z(25)$ =2.87, p=0.003, and the interaction between opponent's overall switch rate and lag-1 switch/no-switch choice, $z(25)$=4.26, p<0.001, but not for opponent's overall switch rate, $z(25)$=1.38, p=0.16 or the player's lag-1 switch/no-switch, $z(25)$=-0.76, p=0.45. Similarly, for the overall rate of winning, we found significant differences for opponents' lag-1 switch/no- switch choice, $z(25)$=2.33, p=0.02, and the interaction between opponents' overall switch rate and the lag-1 switch/no-switch, $z(25)$=2.21, p=0.02, but again not for opponents' overall switch rate, $z(25)$=1.54, p=0.12, and the player's lag-1 switch/no-switch, $z(25)$=-1.18, p=0.23.

Combined, these individual differences results suggest that the degree to which history/context variables are represented in the EEG signal following win feedback, predicts both individuals' reliance on the model of the opponent and their overall competitive success. These relationships are largely absent on post-loss trials. While the absence of a post-win/post-loss difference would have been difficult to interpret for the reasons discussed in the context of the within-subject analysis (i.e., *Table 4*), the fact that we find robust differences here is consistent with the conclusion that following loss-feedback, model-based representations are suppressed and therefore are less relevant for behavior. With its small sample size, this experiment was not designed as an individual differences study and therefore these exploratory results need to be considered with caution. However, confidence in the results is strengthened by the fact that they are largely consistent with the findings from the within-subject PPI analyses.

## Discussion

Our results show that people can use two different choice regimes for selecting their next move in a competitive game. Immediately following a win, participants tended to rely on an internal model of the opponent's behavior and his/her general tendency to switch moves relative to the preceding trial. Following a loss trial, they selected their next move more stochastically and less influenced by the local or global choice context (*Tervo et al., 2014*; *Kolling et al., 2016a*). The demonstration that after loss feedback, model-based selection is largely replaced by a more memory-free, stochastic mode of selection is a theoretically important result.

Past work has characterized behavior in zero-sum game situations as a problem of a trade-off between model-free, win-stay/lose-shift tendencies and choices based on a model of the opponent (e.g., *Lee et al., 2014*). In addition, work with the voluntary task-switching paradigm has suggested that people often tend to repeat action plans that were executed in the immediate past (*Arrington and Logan, 2004*; *Mayr and Bell, 2006*). Indeed, our modeling results show that each of these influences is consistently present in our data. Importantly, the tendency towards model-based choice and the tendency towards stochasticity on post-loss trials, independently predicted individuals' competitive success, and over and above the effect of the known, lower-level biases (see *Tables 2* and *3*).

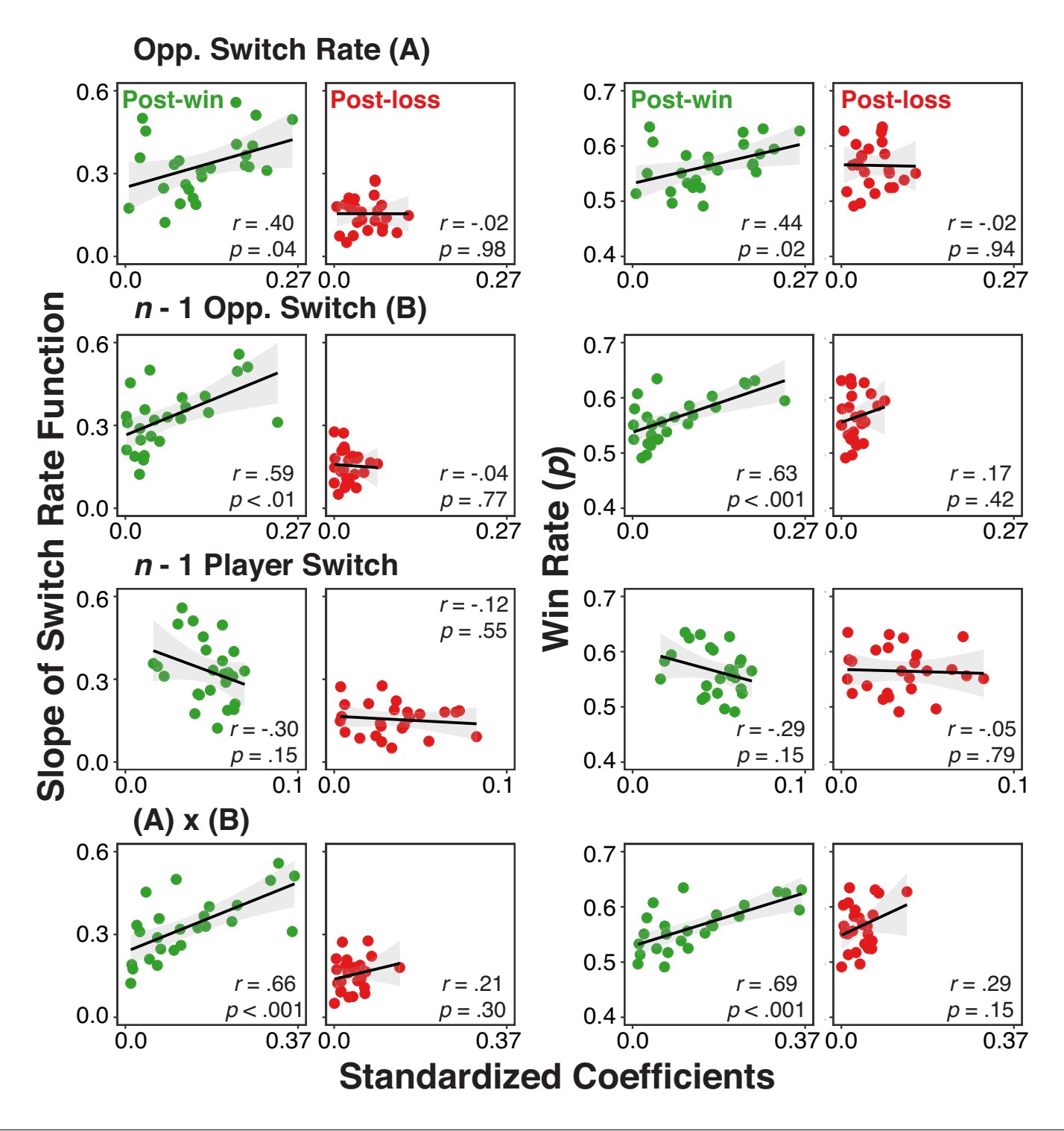

**Figure 6.** Individual difference correlations between neural-level representation of history/context variables and both use of the model and rate of winning. Correlations between individuals' standardized coefficients from the multi-level regression analysis relating the EEG signal to the different history/context variables and 1) their slopes for the switch rate functions (left two columns) or 2) their overall win rate (right two columns) separately for post-win and post-loss conditions. Coefficients were obtained by fitting models with the EEG signals averaged over a 300–700 ms interval of the post-feedback period (the shaded interval in *Figure 5*).

DOI: https://doi.org/10.7554/eLife.48810.021

Our findings are generally consistent with recent research indicating an increase of decision noise—defined as deviations from optimal choice—when exploration is beneficial in a sequential decision situation (*Wilson et al., 2014*). However, it is a novel finding that this stochastic selection regime is turned off and on according to positive versus negative feedback. Also, different than in standard sequential-decision paradigms, in our paradigm subjects chose action rules rather than specific actions (except for Experiment 3), allowing us to separately examine choices and the efficiency of action execution. The fact that the post-loss increase in stochastic choice was *not* accompanied by a consistent increase in action errors or RTs, speaks against the possibility that choice stochasticity results from a general increase in information-processing noise.

The results we report here are consistent of an emerging literature on the asymmetric consequences of positive versus negative feedback (*Sharot and Garrett, 2016*; *Lefebvre et al., 2017*). For example, people appear to update their beliefs to a lesser degree following negative feedback than positive feedback and the neural representation of recent actions is more robust following rewarded than unrewarded trials (*Donahue et al., 2013*; *Wirth et al., 2009*). Particularly important in the current context is the study by *Donahue et al. (2013)*, which had monkeys play a matching pennies game against a computerized opponent, while recording neurons from various regions in the frontal and parietal cortex. Largely consistent with our results, representations of monkeys' recent choices were generally more robust on rewarded trials (the opponents' choices were not examined here), while there was no post-win/post-loss difference in the decoding of the upcoming choices (see *Figure 5—figure supplement 1*).

The existing evidence documents reward-related effects on the representation of the decision maker's specific action choices that *preceded* the positive or negative feedback (choices leading to the outcome). In contrast, we report here that following negative feedback, the representation of broader context information that includes information about the opponent and that is relevant for the *next* action is substantially reduced. Combined, these sets of findings suggest that negative feedback suppresses both learning, and the representation of choice-relevant information that is in principle available to the decision maker. Whether or not these different manifestations of a feedback-related asymmetry share common underlying mechanisms is an important question for further research.

In many competitive situations, a model of the opponent is needed to exploit regularities in the opponent's behavior (*Camerer et al., 2015*; *Lee, 2008*). At the same time, one's own choices need to appear unpredictable to the opponent. The feedback-contingent mix of choice regimes we report here, may be an attempt to meet the opposing demands within the limitations of our cognitive system. By this account, wins signal to the system that the current model is valid and is safe to use. In contrast, losses signal that the current model may be invalid and that alternatives should be explored and/or that there is a danger of being exploited by the opponent. As a result, the current model is temporarily abandoned in favor of stochastic, memory-free choice.

Viewed by itself, the turn toward stochastic choice following losses is an irrational bias. Indeed, our modeling results indicate that the degree to which players switch to stochastic choice after losses, negatively predicts their success in competing against both simulated and actual opponents (see *Tables 2* and *3*). Interestingly, this choice strategy also resembles maladaptive, learned-helplessness patterns that are typically observed across longer time scales and that are often associated with the development of depressive symptoms (*Abramson et al., 1978*; *Maier and Seligman, 2016*). To what degree the trial-by-trial phenomenon examined here and the longer-term, more standard learned-helplessness processes are related is an interesting question for further research. It is however also important to consider the possibility that the loss-contingent switch to stochastic choice is adaptive in certain circumstances. For example, in situations with a greater number of choices or strategies than were available within the fox/rabbit game, a switch to stochastic choice may allow the exploration of neglected regions of the task space (*Cavanagh et al., 2012*). Also, given known constraints on consistent use of model-based selection, the ability to revert to stochastic behavior provides a 'safe default' that at the very least reduces the danger of counter-prediction through a strong opponent.

In this regard, a recent study by *Tervo et al. (2014)* is highly relevant. These authors analyzed the choice behavior of rats playing a matching-pennies games against simulated competitors of varying strength. The animals showed model-based choice behavior against moderately strong competitors, but switched to a stochastic choice regime when facing a strong competitor. The switch in choice

regimes occurred on different time scales in the Tervo et al. study (i.e., several sessions for each competitor) than in the current work (i.e., trial-to-trial). Nevertheless, it is remarkable that a qualitatively similar, failure-contingent switch between choice regimes could be found in both rats and human players.

Tervo et al. also used circuit interruptions in transgenic rats to show that a switch to stochastic choice is controlled via noradrenergic input to the anterior cingulate cortex (ACC), which supposedly suppresses or perturbs ACC-based representations of the current task model. Interestingly, in our Experiment 5, we found that EEG signals registered at mid-frontal electrodes, contained robust information about the opponents' and the players' own strategies following win-feedback. On post-loss trials the EEG signal initially contained information about the opponent's global and local behavior, but this information was all but eliminated by about 400 ms following the feedback signal. This time-course suggests that context information is available in principle, but is quickly suppressed on post-loss trials. Additional analyses indicated that the task-relevant information contained in the EEG signal was indeed relevant for upcoming choice behavior. Feedback-contingent, mid-frontal EEG signals are often thought to originate in the ACC and associated areas (*Kolling et al., 2016a*; *Cavanagh and Frank, 2014*; *Kolling et al., 2016b*). We note though that the earlier-mentioned study by *Donahue et al. (2013)* found reward-related effects on representations of the animals' own recent choices in various frontal and parietal areas, but *not* in the ACC. Aside from obvious differences (i.e., monkeys, rats, and humans, single-cell recordings vs. EEG), one possible reason for these discrepancies is that while Donahue et al. had looked at the representation of the player's own recent history, in our study we found particularly strong feedback effects on the representation of the opponent's behavior. Irrespective of the specific neural-anatomical implementation, the noradrenergic perturbation process identified by Tervo et al., suggests an interesting hypothesis for future research about how in humans, task-relevant representations might be actively suppressed to promote memory-free, stochastic choice (*Nassar et al., 2012*). More generally, this emerging body of evidence provides one possible answer to the fundamental question how a memory-based choice system can produce non-deterministic behavior—namely through temporarily suppressing memory records of the choice context.

## Note added in proof

After acceptance of this manuscript, we became aware of a manuscript by *Hermoso-Mendizabal et al. (2019)*. These authors report a study with rats that both in terms of experimental design and results is remarkably consistent with what we report here. In a serial choice task, rats exploited experimentally induced sequential regularities (i.e., high frequency of repetitions versus alternations) following positive feedback, but temporarily reverted to almost completely stochastic choice behavior following a single, negative feedback trial.

## Materials and methods

### Participants

Subjects were University of Oregon students who participated after giving informed consent in exchange for monetary payment or course credits; Experiment 1: $N$ = 56 (38 female), Experiment 2: $N$ = 40 (28 female), Experiment 3: $N$ = 44 (25 female), Experiment 4a: $N$ = 100 (62 female), Experiment 4b: $N$ = 38 (20 female), Experiment 5: $N$ = 25 (13 female). Four subjects from Experiment 1 and three pairs from Experiment 4a were excluded, because the experimental session could not be completed. The entire study protocol was approved by the University of Oregon's Human Subjects Review Board (Protocol 10272010.016).

### Experiment 1

On each trial of the fox/rabbit game, players observed a circle either on the bottom or the top of a vertically aligned rectangle. They had to choose between one of two rules for responding to the circle location. The 'freeze rule' implied that the circle stayed at the same location and it required participants to press among two keys the one that was compatible with the circle location ('2' and '5' on the number pad), using the right-hand index finger. The 'run rule' implied that the circle moved to the opposite location within the vertical box and participants had to press among a separate set

of vertically aligned keys ('1' and '4' on the number pad) the key that was incompatible with the circle location (*Mayr and Bell, 2006*), using the left-hand index finger. Participants were asked to rest the index finger of each hand between the two relevant keys (e.g., between '1' and '4' for the left index finger) at the beginning of each trial. On a given trial, the fox player won two cent per trial, when both players chose the same rule, whereas the rabbit player won when choices were different. Participants had to respond within a 2000 ms interval and after that interval, they received feedback presented for 200 ms with a smiley face indicating a win trial and a frowny face a loss trial. Both incorrect responses (e.g., a compatible response using the keys for the incompatible rule) or late responses (which were extremely rare) counted as errors. In terms of feedback, all errors were treated in the same way as loss trials, that is a frowny face was presented at the end of the 2000 ms response interval. The inter-trial-interval (ITI) was 300 ms.

Participants initially were exposed to a block of 80 practice trials in order to familiarize them with the response procedures. This block was performed without a competitor, but with the typical 'voluntary switching' instruction that asks subjects to change rules randomly on a trial-by-trial basis. Following practice, participants played the fox/rabbit game for ten different blocks of 80 trials each. Both the switch rate of the opponent varied on a block-by-block basis between 20%, 35%, 50%, 65%, and 80%, and also whether the player had the role of the fox or the rabbit. Except for the switch-rate constraint, the simulated opponent's choices were completely random. The ten different combinations of opponent switch rate and player role were randomly distributed across blocks. Participants were instructed that the different simulated players represented common strategies that one might find in human players. At the beginning of each block, participants were notified that they would be facing a new, simulated opponent, and whether they played the role of the fox or the rabbit, but received no instruction about the specific strategies.

This and the following experiments were programmed in Matlab (Mathworks) using the Psychophysics Toolbox (*Brainard, 1997*) and presented on a 17-inch CRT monitor (refresh rate: 60 Hz) at a viewing distance of 100 cm.

## Experiment 2
This experiment was identical to Experiment 1 in all aspects, only that here the ITI varied between 300 ms and 1000 ms randomly, on a trial-by-trial basis.

## Experiment 3
This experiment was again identical to Experiment 1, only that here the choice between two different response rules was replaced by a simple choice between two different key-press responses. Each trial was initiated by a circle appearing at the center of the vertically arranged stimulus rectangle. Using the vertically arranged '2' or '5' keys, participants had to shift the circle up or down within the rectangle. Matching moves between opponents implied a win for the fox and a loss for the rabbit player.

## Experiment 4
In Experiment 4a participants were paired into fox/rabbit dyads and played in real-time on two computers within the same room, but without opportunity for direct communication. Here, participants played 7 blocks of 80 trials each, and stayed within the same role throughout the experiment. All other aspects were identical to Experiment 1. Experiment 4b served as a non-competition control experiment. Here, participants were given the standard instruction for the voluntary task-switching paradigm, namely to select tasks as randomly as possible ("simulating a series of coin tosses"). Also, trial-by-trial wins and losses were completely random and participants were informed that this was the case. Otherwise, this experiment was identical to Experiment 4a.

## Experiment 5
This experiment was again identical to Experiment 1, but optimized towards EEG recording. For this purpose, we used only three simulated opponents (switch rates of 25%, 50% and 75%) across 24 blocks of 80 trials each (i.e., 4 repetitions of 3 opponent strategies x fox/rabbit roles). The ITI was 700 ms to allow assessment of feedback-related EEG activity.

## History analyses

To evaluate the predictability of the current choice by the recent choice history, we fitted multilevel logistic regression models predicting the switch vs. repeat choice by the player's and opponent's switch history from $n$-3, $n$-2, and $n$-1 trials, the overall switch probability of the opponent, whether trial $n$-1 was a win or a loss trial, and the interactions between win/loss and all history/context variables (i.e., 15 predictors in total). We estimated both fixed and random effects of all predictors. For Experiment 4, the model had three levels in which trials were nested within players, which in turn were nested in dyads. For the other experiments, models included only the first two levels. The signature of model-based selection is that predictors representing the opponent's switch rate (e.g., the overall switch probability and opponent's switch history) is positively related to the player's switch probability on post-win trials and negatively on post-loss trials. The main prediction we wanted to test was that following post-win trials the predictive relationship is stronger than following post-loss trials. Therefore, we examined the interaction between the post-win/loss contrast and each opponent-related predictor after reversing the label of the predictor for post-loss trials (e.g., $n$–one opponent 'switch' is relabeled as 'repeat', 80% overall switch rate becomes 20%). This allowed us to test the difference in the strength of the relationship, while ignoring the direction of the relationship. We had no a-priori prediction about the direction of the relationship between previous switch/repeat choices and the trial $n$ switch/repeat choice. Nevertheless, for a conservative test of post-win/loss differences we again reversed the post-loss label in each case where there was an empirical flip in sign between post-loss and post-win coefficients. We also present in *Figure 4—figure supplement 2* the results without reversing labels.

## EEG recordings and analysis

In Experiment 5, Electroencephalographic (EEG) activity was recorded from 20 tin electrodes held in place by an elastic cap (Electrocap International) using the International 10/20 system. The F3, Fz, F4, T3, C3, CZ, C4, T4, P3, PZ, P4, T5, T6, O1, and O2 of the 10/20 system were used along with five nonstandard sites: OL midway between T5 and O1; OR midway between T6 and O2; PO3 midway between P3 and OL; PO4 midway between P4 and OR; and POz midway between PO3 and PO4. The left-mastoid was used as reference for all recording sites. Data were re-referenced off-line to the average of all scalp electrodes. Electrodes placed ~1 cm to the left and right of the external canthi of each eye recorded horizontal electrooculogram (EOG) to measure horizontal saccades. To detect blinks, vertical EOG was recorded from an electrode placed beneath the left eye and reference to the left mastoid. The EEG and EOG were amplified with an SA Instrumentation amplifier with a bandpass of 0.01–80 Hz and were digitized at 250 Hz in LabView 6.1 running on a PC. We used the Signal Processing and EEGLAB (*Delorme and Makeig, 2004*) toolboxes for EEG processing in MATLAB. Trials including blinks (>60 μv, window size = 200 ms, window step = 50 ms), large eye movements (>1°, window size = 200 ms, window step = 10 ms), and blocking of signals (range = −0.01 μv to 0.01 μv, window size = 200 ms) were rejected excluded from further analysis.

Single-trial EEG signals were segmented into 1250 ms epochs starting from 200 ms before the onset of feedback. Thus, each epoch included 700 ms post-feedback periods and the initial 250 ms intervals of the next trials. Each electrode's EEG signal was also pre-whitened by linear and quadratic trends across experimental trials and blocks. After baselining signals with data from the initial, 200 ms interval, EEG activity from electrodes Fz and Cz, was averaged. These electrodes were selected based on previous studies reporting a robust interaction between the feedback and the probability context during reinforcement learning (*Cohen et al., 2007*). The resulting signal was regressed via multilevel modeling with two levels (i.e., trials nested within participants) on context variables, as described in the Results section. For illustrative purposes, this was done on a time-point by time-point basis (see *Figure 5*). To conduct statistical tests of the post-win versus post-loss regression coefficients for the psychophysiological interaction analysis predicting choices, for the individual differences (*Figure 6*), and for the topographic maps (*Figure 5a*), we averaged the EEG signal for an a-priori defined 300–700 ms interval from the onset of feedback up to the beginning of the next trial. This interval is based on the typical time-course of feedback effects reported in the literature (*Cohen et al., 2007*). The difference between post-win/loss models was tested in the same manner as in the multilevel model for history effects, namely by inverting predictors of opponents' history/

context for post-loss trials (see section *Analytic Strategy for Testing Main Prediction* and *History Effects Analysis*).

## Acknowledgements

We thank Chihoko Hayashi, Katelyn Occhipinti, Joshua Karpf, Ali Byer, Christine Manalansan for their assistance with data collection.

## Additional information

### Funding

| Funder | Grant reference number | Author |
|---|---|---|
| National Institutes of Health | R01 AG037564- 01A1 | Ulrich Mayr |

The funders had no role in study design, data collection and interpretation, or the decision to submit the work for publication.

### Author contributions

Atsushi Kikumoto, Data curation, Software, Validation, Investigation, Visualization, Methodology, Writing—original draft, Writing—review and editing, Formal analaysis; Ulrich Mayr, Conceptualization, Formal analysis, Supervision, Funding acquisition, Investigation, Methodology, Writing—original draft, Project administration

### Author ORCIDs

Atsushi Kikumoto (iD) https://orcid.org/0000-0002-2179-2700
Ulrich Mayr (iD) https://orcid.org/0000-0002-7512-4556

### Ethics

Human subjects: The entire study protocol and consent forms were approved by the University of Oregon's Human Subjects Review Board (Protocol 10272010.016).

### Decision letter and Author response

Decision letter https://doi.org/10.7554/eLife.48810.028
Author response https://doi.org/10.7554/eLife.48810.029

## Additional files

### Supplementary files

• Transparent reporting form DOI: https://doi.org/10.7554/eLife.48810.022

### Data availability

Data and analyses are available through OSF (https://osf.io/j6beq/). Specifically, the repository contains for each of the five experiments, all trial-by-trial data files, as well as R codes to conduct the reported analyses. For Experiment 5, we also include all relevant EEG data and analyses codes.

The following dataset was generated:

| Author(s) | Year | Dataset title | Dataset URL | Database and Identifier |
|---|---|---|---|---|
| Kikumoto A, Mayr U | 2019 | Balancing model-based and memory-free action selection under competitive pressure | https://osf.io/j6beq/ | Open Science Foundation, j6beq |

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
