## [Decision Letter]

Thank you for submitting your article "Balancing model-based and memory-free action selection under competitive pressure" for consideration by *eLife*. Your article has been reviewed by three peer reviewers, and the evaluation has been overseen by a Reviewing Editor and Michael Frank as the Senior Editor. The following individuals involved in review of your submission have agreed to reveal their identity: Alex C Kwan (Reviewer #1); Christopher H Donahue (Reviewer #2).

The reviewers have discussed the reviews with one another and the Reviewing Editor has drafted this decision to help you prepare a revised submission.

Summary:

Authors conducted a set of experiments using the variants of matching pennies game to understand how multiple decision and learning strategies might be arbitrated and influenced by the previous choice outcomes. They found that the strategy changed depending on whether the previous choice produced positive or negative outcomes. In addition, this was also reflected in the EEG signals recorded from the mid-frontal regions. A strength of this study is how by systematically varying the simulated opponents' switching probability the authors were able to reveal convincing behavioral evidence that a more global model-based strategy was used more strongly after wins vs. losses. The conclusions are well supported by the data.

Essential revisions:

1) The writing should be improved and some additional details should be provided in the Results. For example:

1a) The different experiments (i.e., Experiments 1, 2, 3, 4a, 4b, and 5) need to be explained better. In the main text, the reasons that motivate each Experiment are provided, however, there lack details to understand what is changed in the actual task design. Some additional explanation in the main text would help the readers so they don't have to refer to the Materials and methods each time. Then, in the Materials and methods section, the description of the Experiments is poor. Right now, the authors chose to describe a step and say it applies to Experiments 1, 3, 4, and then another step and say it applies to Experiment 2, 5, etc. A clearer approach would be to put each Experiment in its own section in the Materials and methods. For example, for the sake of clarity, when describing Experiment 2, some steps can be repeatedly describe and then also explain how it differs from Experiment 1. As a result, it is quite difficult to know exactly what the players were doing in the main part of the manuscript, so Experiment 3 (where they switch to traditional matching pennies) was confusing without reading through the Materials and methods. The description in "overview" just sounds like traditional matching pennies, but there is no indication there that every experiment except Experiment 3 used four response keys.

The authors should explain better why exactly the paradigm was altered in this manner. The authors' link it to "voluntary task switching" paradigms, but that logical connection and the implications are unclear. The logic surrounding this seems to be spelled out as allowing the distinction between "two potential sources of stochasticity", but this was still hard to follow. Was this added level of executive demand expected to impede the employment of model-based strategies on loss trials, selectively? Another justification for this choice of paradigm is that it "recreates executive control demands" in competitive scenarios. But what is such an example of an executive control demand that is selectively present for switching rather than staying with a choice, that is not capture by mapping actions onto different keys?

1b) From Experiment 1, for 80% opponent switch rate, the win rate is relatively low according to Figure 2—figure supplement 2. This is consistent and related to the fact that although subject adapts a model-based like strategy, the subjects still switch only ~60% given the opponent is switching at 80%. The subjects should switch at a higher rate post-win to gain further advantage. However the subject seems to be better at gaining this leverage at the low end of opponent switch rate. What may cause this asymmetry?

1c) The authors should clarify the relationship between choice inputs and noise, and how that can be dissociated based on response time and error rates. Currently it is taken as evidence that RT and error rates do not change that it is the choice inputs that have been reduced. But presumably noise is always present in the brain's decision-making system. So, any reduction in choice input means that the relative influence of noise is increased which would impact response time and noise. The authors are making some assumptions, and those should be stated in this interpretation.

1d) The Materials and methods should contain more details about how the computer opponent is implemented. Relatedly, for the computer opponent, it seems that the current implementation means that the computer does not consider the choice and reward history. What if the computer opponent takes some of that into account for its current decision? How would that influence the use of model-based versus memory-free strategy?

2) Another weakness is in regards to their EEG analysis which seems preliminary. The regression model used in the analysis seems to be missing some important factors related to sensory and motor information, but the interpretation of the psychophysical interaction analysis relies so heavily on this step. It is important to control for sensory and motor information when analyzing the EEG data. The authors should include regressors related to the position of the circle as well as the motor response (up vs. down). It may also be useful to show the time course of these regressors aligned to stimulus onset and the motor response.

3) There are some questions about the PPI analysis. In Figure 5, the time course of the regression coefficient for the interaction term looks to be almost identical to the one for the opponent's switch rate. Is it necessary to include this interaction term? Would AxC or even an interaction between A and a random vector look similar? In addition, the only data supporting the conclusion that the asymmetries in feedback representations are related to the next action vs. preceding actions appears to be the PPI analysis (Discussion, fourth paragraph), but this is not very convincing and the conclusion needs more support. Although there are important differences in task design which may make comparison difficult, results from recordings in non-human primates (Donahue, Seo, and Lee, 2013, Figure 5) clearly demonstrate the opposite.

4) Regarding Figure 4, is there an explanation for why some of the coefficients related to longer trial lags (n-3) do not decay to zero? Is there something important in the behavior that this model is not capturing?

5) Equation 1 is confusing. Does it miss a sign? For example, if we assume that the opponent behaves randomly (setting *os*=0), then including a positive *pe* term would actually lead to more switching in the model as written.

6) The manuscript does not appear to include the definition of an error. Are they omissions? Incorrect button presses?

---

## [Author Response]

Essential revisions:1) The writing should be improved and some additional details should be provided in the Results. For example:1a) The different experiments (i.e., Experiments 1, 2, 3, 4a, 4b, and 5) need to be explained better. In the main text, the reasons that motivate each Experiment are provided, however, there lack details to understand what is changed in the actual task design. Some additional explanation in the main text would help the readers so they don't have to refer to the Materials and methods each time. Then, in the Materials and methods section, the description of the Experiments is poor. Right now, the authors chose to describe a step and say it applies to Experiments 1, 3, 4, and then another step and say it applies to Experiment 2, 5, etc. A clearer approach would be to put each Experiment in its own section in the Materials and methods. For example, for the sake of clarity, when describing Experiment 2, some steps can be repeatedly describe and then also explain how it differs from Experiment 1. As a result, it is quite difficult to know exactly what the players were doing in the main part of the manuscript, so Experiment 3 (where they switch to traditional matching pennies) was confusing without reading through the Materials and methods. The description in "overview" just sounds like traditional matching pennies, but there is no indication there that every experiment except Experiment 3 used four response keys.

We followed all of these suggestions: We now go through each experiment step by step in the Materials and methods and the Results section, and we also try to more clearly explain the response format.

The authors should explain better why exactly the paradigm was altered in this manner. The authors' link it to "voluntary task switching" paradigms, but that logical connection and the implications are unclear. The logic surrounding this seems to be spelled out as allowing the distinction between "two potential sources of stochasticity", but this was still hard to follow. Was this added level of executive demand expected to impede the employment of model-based strategies on loss trials, selectively? Another justification for this choice of paradigm is that it "recreates executive control demands" in competitive scenarios. But what is such an example of an executive control demand that is selectively present for switching rather than staying with a choice, that is not capture by mapping actions onto different keys?

We now try to more clearly articulate the reason for using this rule-selection version of the matching-pennies game (Results subsection “Overview”), in particular describing in detail how we can distinguish between choices and action errors (see also point 1c). However, we also emphasize that our results generalize to a standard version of the matching-pennies game (Experiment 3). Originally, the specific executive-control demands of choosing to switch between different rules were an additional motivation for choosing the rule-selection paradigm. However, we now de-emphasize this aspect as it is somewhat peripheral to our main conclusions.

1b) From Experiment 1, for 80% opponent switch rate, the win rate is relatively low according to Figure 2—figure supplement 2. This is consistent and related to the fact that although subject adapts a model-based like strategy, the subjects still switch only ~60% given the opponent is switching at 80%. The subjects should switch at a higher rate post-win to gain further advantage. However the subject seems to be better at gaining this leverage at the low end of opponent switch rate. What may cause this asymmetry?

As the modeling results indicate, overall choice results are not just driven by model-based and stochastic choices, but also by perseveratory and win-stay/lose-shift strategies, which both work specifically against the expression of high switch rates (and in particular in the case of post-win trials). We now try to articulate more clearly that we believe that these “lower-level” biases are responsible for the observed asymmetry (subsection “Choice behavior with simulated opponents”, second paragraph).

1c) The authors should clarify the relationship between choice inputs and noise, and how that can be dissociated based on response time and error rates. Currently it is taken as evidence that RT and error rates do not change that it is the choice inputs that have been reduced. But presumably noise is always present in the brain's decision-making system. So, any reduction in choice input means that the relative influence of noise is increased which would impact response time and noise. The authors are making some assumptions, and those should be stated in this interpretation.

We now try to more clearly articulate our assumptions (e.g., Results, subsection “Overview”). The point is well-taken that it may still be the information-processing noise (that is indicated by response error rates) that drives stochastic behavior in the absence of an adequate context representation. However, even if this were the case, we believe that we can still argue that it is the dampening of the context representation following losses that would allow information processing noise to obtain control over choice behavior.

1d) The Materials and methods should contain more details about how the computer opponent is implemented. Relatedly, for the computer opponent, it seems that the current implementation means that the computer does not consider the choice and reward history. What if the computer opponent takes some of that into account for its current decision? How would that influence the use of model-based versus memory-free strategy?

We now emphasize that the simulated opponents’ choices were random, except for the switch-rate constraint (Results, subsection “Overview” and Materials and methods, subsection “Experiment 1”).

2) Another weakness is in regards to their EEG analysis which seems preliminary. The regression model used in the analysis seems to be missing some important factors related to sensory and motor information, but the interpretation of the psychophysical interaction analysis relies so heavily on this step. It is important to control for sensory and motor information when analyzing the EEG data. The authors should include regressors related to the position of the circle as well as the motor response (up vs. down). It may also be useful to show the time course of these regressors aligned to stimulus onset and the motor response.

Thank you for this suggestion. We now present these additional analyses (Figure 5—figure supplement 1). Inclusion of these predictors does not change the pattern of results.

3) There are some questions about the PPI analysis. In Figure 5, the time course of the regression coefficient for the interaction term looks to be almost identical to the one for the opponent's switch rate. Is it necessary to include this interaction term. Would AxC or even an interaction between A and a random vector look similar? In addition, the only data supporting the conclusion that the asymmetries in feedback representations are related to the next action vs. preceding actions appears to be the PPI analysis (Discussion, fourth paragraph), but this is not very convincing and the conclusion needs more support. Although there are important differences in task design which may make comparison difficult, results from recordings in non-human primates (Donahue, Seo, and Lee, 2013, Figure 5) clearly demonstrate the opposite.

Regarding the interaction term:

Author response table 1 shows the following:

1) The results of the original analysis.

2) An analysis that excludes the interaction term and shows that inclusion of the interaction term is not necessary to obtain the win/loss difference for the opponent’s switch rate and lag-1 switch, but strengthens the pattern. We do want to stress that including this interaction term was theoretically motivated as it reflects the prediction that the neural signals are particularly sensitive to discrepancies between overall expectations (driven by global switch rate) and local events (most recent switch/no-switch).

3) An analysis that adds to the original analysis also the interaction between player lag-1 switch and opponent switch rate. This analysis shows very similar effects for both interaction terms. Thus the player’s last switch/no-switch choice affects the EEG signal differentially depending on the opponent switch rate – but only following win trials. While we had not predicted this pattern a priori, it is consistent with the overall conclusion that history/context effects are less strongly represented after wins than losses. Given that we had not predicted this effect, we would like to stick to our original analysis in the paper.

We also conducted the suggested control analyses using the interaction with a random vector – this produced no significant effect for the interaction, and no change in the pattern of the remaining predictors compared to the analysis with no interaction.

**Author response table 1. resptable1:** Absolute coefficients from the MLM regression predicting trial-to-trial EEG signals with additional control predictors.

		Post-win	Post-loss
Model		*b*	*se*	*t*-value	*b*	*se*	*t*-value
Original	Opponent Switch Rate(A)	.120	.00	11.04	-.033	.012	-2.80
	*n*-1 Opponent Switch (B)	.062	.018	5.66	<-.001	.008	-0.05
	*n*-1 Player Switch	.048	.007	6.32	.003	.008	4.06
	(A) x (B)	-.134	.010	-12.11	.028	.012	2.15
No interaction	Opponent Switch Rate	.038	.014	2.74	-.011	.015	-.743
	*n*-1 Opponent Switch	.039	.011	3.66	<.001	.010	.083
	*n*-1Player Switch	.049	.009	5.24	.010	.009	3.27
*n*-1 Player switch	Opponent Switch Rate(A)	.184	.162	14.68	-.265	.015	-1.79
	*n*-1 Opponent Switch (B)	.049	.013	5.61	<.001	.008	.076
	*n*-1 Player Switch (C)	.041	.001	5.96	.033	.008	4.10
	(A) x (B)	-.110	.011	-10.47	.025	.012	2.05
	(A) x (C)	-.102	<.001	-10.93	.001	.011	-.89

Note. Shown are the unstandardized regression coefficients (*b*), the standard error around the coefficients (*se*), and the associated *z* values. In order to allow comparison of coefficients cross models, only random intercepts were estimated for all models as for the *N*-1 Player Switch model with full random effects did not converge. The full models for the original and the no-interaction model show the same pattern of results, albeit overall lower coefficients.

Regarding the interpretation of asymmetries and the Donahue et al. paper:

We agree that the statement referenced (in original document) is not sufficiently clear. In fact, our data provide no strong support for the conclusion that “people make less use of relevant representations as they choose the next action”. What we can say with some confidence though is that (a) history/context is less strongly represented following loss trials, and that (b) trial-to-trial variations in the strength of this representation are predictive of choice behavior. We clarify this aspect now (subsection “Neural evidence for memory-free choice following losses: Experiment 5”, seventh paragraph).

Thank you also for bringing the highly relevant Donahue et al. paper to our attention, which we now refer to in some detail (e.g., Discussion, fourth and last paragraphs). The results in this paper and our findings are quite similar with regard to the fact that past choice behavior is less represented following negative than following positive feedback (except in the ACC, which we now also note in the Discussion). It is however important to acknowledge that this paper analyzed the representation of the player’s past behavior, not of the opponent’s overall strategies or recent choices as we do in our analyses. Where Donahue et al.’s results seem to diverge from ours is that they found no consistent reward-dependent representation of the animal’s upcoming choice. The main point of our PPI analyses was to demonstrate that the context representations tracked through EEG (and that show a clear feedback asymmetry) are actually relevant for choice behavior. Therefore, we show that trial-to-trial variation in the representation of the context variables predicts upcoming choices. The main point here is *not* that this relationship itself is reduced following losses, as we discuss in the subsection “Neural evidence for memory-free choice following losses: Experiment 5”, seventh paragraph (see also the last paragraph of the aforementioned subsection).

In contrast to our analysis focusing on the degree to which context is represented as a predictor of upcoming choices, Donahue et al. report that the upcoming choice can be predicted by neural activity and that this predictive relationship is not reduced after loss trials. Note, that we also report predictability of the upcoming choice through the EEG data in a control analysis (Figure 5—figure supplement 1). As Donahue et al., here we do not find a marked difference between post-win and post-loss trials, a result that we had not specifically highlighted in the previous version, but now do (subsection “Neural evidence for memory-free choice following losses: Experiment 5”, fourth paragraph, Discussion, fourth paragraph, and Figure 5—figure supplement 1 legend).

To summarize, there are important similarities in the results of these studies as both show that some aspects of the choice history/context are less strongly represented following losses. We believe that the differences are largely due to different analyses (e.g., the direct decoding of upcoming choice vs. strength of decoding of history/context as predictor of upcoming choice).

Also, we believe that our PPI results are bolstered by the fact that the strength of history/context representation not only predicts choice within individuals (i.e., across trials), but also predicts individual differences in choice behavior and competitive success (Figure 6).

4) Regarding Figure 4, is there an explanation for why some of the coefficients related to longer trial lags (n-3) do not decay to zero? Is there something important in the behavior that this model is not capturing?

This is a good point. Large coefficients for farther back trials are present mainly for the player’s choice history. Note, that for the opponent, we had included the global switch rate, which varies on a block by block manner, but we had not included a similar variable for the player. Thus, we thought it is possible that the more remote history of the player captures larger-scale (e.g., block-by-block) variations in the player’s switch rate. In fact, when we include the blockwise switch-rate of the player as additional variable, the coefficients for longer lags largely disappear (as an example, Author response image 1 shows the original and pattern of coefficients with players’ blockwise switch rate included for Experiment 1). We do not want to include this variable in the main analysis, as it is highly correlated with the opponents’ switch rate (which one would expect when people behave in a model-based manner) and the collinearity complicates the interpretation of coefficients (note, though that the coefficients for opponent’s history are not affected). We also note, that the qualitative pattern of results regarding the opponent-related predictors do not change when we drop all player history predictors from the model.

5) Equation 1 is confusing. Does it miss a sign? For example, if we assume that the opponent behaves randomly (setting os=0), then including a positive pe term would actually lead to more switching in the model as written.

Thank you for catching this mistake. The equation now reflects the model that was actually fitted to the data.

6) The manuscript does not appear to include the definition of an error. Are they omissions? Incorrect button presses?

We now clearly define errors and how they were treated (Results, subsection “Overview” and Materials and methods, subsection “Experiment 1”).